# Web-Based Integer Programming Decision Support System for Walnut Processing Planning: The MeliFen Case

**Carlos F. Brunner-Parra** [1], **Luis A. Croquevielle-Rendic** [2], **Carlos A. Monardes-Concha** [2,3,*], **Bryan A. Urra-Calfuñir** [3], **Elbio L. Avanzini** [2] and **Tomás Correa-Vial** [4]

1   Department of Transport Engineering and Logistics, Pontificia Universidad Católica de Chile, Santiago 7820436, Chile; cfbrunner@uc.cl
2   Industrial and Systems Engineering Department, Pontificia Universidad Católica de Chile, Santiago 7820436, Chile; lacroquevielle@uc.cl (L.A.C.-R.); elavanzini@uc.cl (E.L.A.)
3   School of Engineering, Universidad Católica del Norte, Coquimbo 1781421, Chile; bryan.urra@alumnos.ucn.cl
4   General Management Department, MeliFen, Paine 9540000, Chile; t.correa@melifen.com
*   Correspondence: cmonardes@ucn.cl; Tel.: +56-51-220-5997

**Abstract:** Chile is among the largest walnut producers and exporters globally, thanks to a favorable nut growth and production environment. Despite an increasingly competitive market, the literature offers little scientific advice regarding decision support systems (DSSs) for the nut sector. In particular, the literature does not present optimization approaches to support decision-making in walnut supply chain management, especially the processing planning. This work provides a DSS that allows the exporter to plan walnut processing decisions taking into account the quality of the raw material, such as size, color, variety, and external and internal defects, in order to maximize the benefits of the business. To formalize the problem, an integer programming model is proposed. The DSS was implemented via a web application for MeliFen, a walnut exporter located near Santiago, Chile. A comparative analysis of the last two years revealed that MeliFen increased its profit by approximately 9.8% using this tool. We also suggest other uses that this DSS provides, besides profit maximization.

**Keywords:** walnut processing planning; integer programming; decision support system; web application

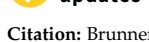



## 1. Introduction

Walnuts are one of the most traded nuts globally today, mainly due to their nutritional value in the form of high protein and vitamin content [1]. Because of the high demand, walnut production has continuously expanded, reaching a 2.3 million t in-shell basis in 2020, with China, the United States, and Chile accounting for over 80% of the total [2].

The work of producers has been fundamental to achieving sustainable development, healthy production, and industry growth. However, they handle small volumes, and their sales are generally limited to domestic customers. For this reason, the role of walnut exporters has been fundamental in making fruit available around the world and marketing and selling local fruit to international customers. Therefore, the exporter must have a clear assessment of the production capacity of its suppliers, keep delivery periods as short as possible, and comply with the quality and the standards that the customer expects. Otherwise, the reputation of the company and even the country may be damaged.

The largest percentage of Chilean exporters of walnut focus on sales of in-shell walnuts since no post-processing of the fruit is required. On the other hand, a smaller fraction deals with a broader range of products: in-shell, machine cracked, and hand-cracked walnuts. MeliFen is a Chilean exporter of walnut that is part of this group. They started exporting in 2015 to different countries in the three main categories: hand-cracked shelled walnuts to Europe, the United Arab Emirates, Qatar, and Russia; machine-cracked shelled walnuts, mainly to Brazil and Germany; in-shell walnuts to all markets in different configurations [3].

MeliFen has grown considerably in the last six years due to the global increase in demand for walnuts. However, global supply has also increased, to the point that China, the largest consumer and producer of walnuts, now exports the fruit, which was previously only used for domestic consumption [2]. Because of the increased competition, the company has sought to strengthen the quality standard to differentiate itself. Therefore, it has been necessary to improve both production planning and decision support.

The walnuts that come from farmers in the same delivery are identified as a unique arriving batch (AB), denoted mathematically as $\ell$. In each AB, there are different qualities, characterized by some key features, which determine how the AB can be processed and sold. Among these factors, some of the most important are the units per kilogram, external and internal defects of the walnuts, shape, size, category, variety, and color. Each AB is analyzed at the moment of reception, providing to MeliFen a functional characterization of the material. If an arriving batch is sub-optimally processed, more kilograms must be destined to maintain export standards and demands, causing higher process costs, overtime costs, shipping costs, a lower return for the producer, and even damaging the corporate image.

To solve this problem, we formulated the decision-making through a mathematical programming model. Then, we implemented this approach via a web-based integer programming (IP) decision support system (DSS). The IP model aims at maximizing the return of the walnut exporter, while the DSS provides dashboards with key performance indicators (KPIs) that help the company make flexible, valuable, and optimal operational decisions to utilize resources efficiently.

The DSS procedure is divided into a two-step process. The first step is a preprocessing phase that calculates the parameters that depend on the conditions of the nut sector and the specific commercial conditions of the producers and the exporter. The second stage is the optimization model, which decides the best way to sell the walnut batch based on the parameters previously calculated. In addition, the optimization model is flexible, allowing the management of several parameters that facilitate the simulation of different possible scenarios through the web application. The existing scientific literature on DSSs in agriculture is extensive and currently focuses on simulation tools [4–8] and specific optimization models and methods [9–14] that have improved the efficiency of agricultural activities significantly, thanks to the use of current technologies such as the Internet of Things, Big Data, artificial intelligence, cloud computing, remote sensing, etc. [15]. Applications of these technologies have enabled online data collection using remote sensing techniques [16], Geographic Information Systems [17,18], sensors [19–21], or uploaded directly by users to web-based services [22] or cloud-based services [23].

Although the number of articles that present DSSs in agriculture is increasing, to the best of our knowledge, there are no studies in the literature addressing DSSs in the nut sector. Hence, the main contribution of this paper to fill the research gap in the existing literature is the development of a web-based integer programming DSS for the walnut industry and its practical application with a Chilean exporter, with the aims of (i) solving walnut AB processing planning to maximize the profit of the exporter and (ii) providing an efficient technological tool (web application) to support the tactical/operational decisions of the exporters by visualizing the optimal plan.

The paper is structured as follows. In Section 2, a brief literature review is conducted. In Section 3, we explain the problem statement. The IP model details for profit maximization and a preprocessing procedure are developed in Section 4. Section 5 considers implementing the optimization solution and the use and operation of the DSS. In Section 6, the model validation, computational results, and sensitivity analysis of the potential benefits are presented. Finally, some conclusions and future work are presented in Section 7.

## 2. Literature Review

DSSs are computer-based systems for supporting complex managerial decision-making processes [24,25]. They have been studied for more than 50 y, and today, they are more

necessary due to growing complexity and uncertainty in many decision-making processes. A model-based DSS is a particular type of DSS, which is designed to be used as a what-if analysis for a non-technical user [26]. Even though a model-based DSS is created using one or more models to suggest decisions to users (usually managers), it should not be focused on only setting parameters and solving equations. The design's sight must be more comprehensive, incorporating the end-user in the implementation and knowing its limitations [25].

For many years, the agriculture industry has seen in the DSSs an opportunity to make more efficient decision-making processes. Unfortunately, many of the designed agricultural DSSs (aDSSs) only have remained in the initial phase: a pilot [27]. Many studies have been conducted for almost 30 y ago, trying to overcome this reality. Greer et al. [28] developed a method to adapt explanations from an aDSS's results to the final user. That method was tested through the EXPLAIN system, which traduces complex simulations results into simple reports for farmers seeking their acceptance in making decisions. McCown [29] explained that the acceptance problem comes from two radical farming management points of view: technological vs. social. He emphasized the social dimension as the key to success in DSS development and that importance is positively correlated with the model's complexity. Other researchers, such as Matthews et al. [27], Jakku and Thorburn [30], Rose et al. [31], and Mir and Padma [32], also agree with the social dimension, suggesting that to reach a successful implementation and adaptation of a DSS, the active participation of end-users during its development is required.

The advancement of technology applications has reduced the costs and times associated with collecting information through user interaction and the regularization of data online, which use various platforms that utilize cloud-based or web-based services. Rupnik et al. [23] developed a cloud-based aDSS that integrates different information systems and continuously updates the data in the cloud. Weisong et al. [22] developed a suitability evaluation system for Chinese table grape production based on WebGIS. In addition to integrating economic benefits, climate, soil, and vineyard location, certain users can upload production information to make decisions from the results delivered by the system. In the agriculture industry, decision support web services are becoming increasingly popular; however, they require collaboration among stakeholders for adoption and long-term use of the technologies [17].

It is known that designing an aDSS is quite complicated, particularly for resource planning; the approaches to this type of problem have been unable and inefficient to control its complexity. This process involves task scheduling, investment analysis, machinery selection, cost analysis, and other aspects of agricultural production [33]. In this way, farmers and their workers can benefit greatly, using an efficient aDSS focused on evaluating and improving the distribution and allocation of resources for the studied process, and taking into account that uncertainty is a factor that can often cause significant problems [34].

Throughout history, many specialized tools in agricultural systems have been developed, contributing to a wide range of disciplines [35]. A wide variety of aDSS applications have been reported in the literature. In a general setting, the AgriSupport II project, documented in Recio et al. [9], consisted of set tools' implementation to design and build DSSs for agriculture advising, supporting field operation planning. In the work of Navarro-Hellín et al. [21], they made an automatic Smart Irrigation DSS, called SIDSS, to optimize the irrigation process in agriculture. This system uses different sensors to integrate soil and climatic measures to feed machine learning algorithms to predict the weekly water needs.

The system is empowered by the precision agriculture context, using various tools to analyze and predict the proper use of available resources. Zhai et al. [15] presented a new survey of aDSSs based on available technology in Agriculture 4.0. This research gathered the current data, methodologies, models, and tools, analyzing and processing the data, transforming them into valuable information, and making agriculture decisions. After thirteen aDSSs were studied, the researchers highlighted the challenges to creating a helpful aDSS in the Agriculture 4.0 context: (1) a simple user interface, (2) improved

functionalities, (3) adapting capacity to make new decisions facing new scenarios, (4) making robust decisions, (5) incorporating expertise to support decision-making, (6) including prediction models, and (7) giving feedback from historical decisions' reports.

Regarding optimization of decision support in the nut sector, the literature rarely focuses on mathematical programming models that solve real-world problems. Among the few cases is the study by Gilani and Sahebi [36], who presented a mixed-integer linear programming (MILP) model that maximizes the total benefits and minimizes the environmental effect of the pistachio supply chain, with a robust possibilistic programming approach to deal with the inherent uncertainties. Similarly, Salehi-Amiri et al. [37] developed a MILP that minimizes the total cost of the sustainable closed-loop supply chain network for the walnut industry. In addition, they employed famous and also recently developed meta-heuristic algorithms and their hybrids to verify the model and reach the best results.

We present a mathematical model-based DSS that solves resource planning in the nut sector, precisely walnut batch processing optimization for a real-world problem. Following the suggestions of Zhai et al. [15], we derived the computational package in a web user interface in which decision-makers can manage their information online for a more efficient and straightforward process. The visualization of the optimal plan can give valuable managerial insights, for example planning investments for improving the better/worst production processes, thus solving the communication between the exporter and the agricultural producer. Furthermore, the technology tool helps the exporter's accountability and justifies why this way should sell the product.

## 3. Problem Statement

Walnut producers are generally focused on the growth and harvest of the fruit. The commercialization of the product is often referred to as exporters that are in direct contact with buyers. The objective of the exporter is to maximize the profit of their producers, so they can keep a good relationship with them and obtain good commissions in the process. The return for the producers is calculated after subtracting processing and export costs from the selling price. The exporter's return is obtained through a flat commission over the selling price charged to each arriving batch, and the industrialization services benefit.

To stand out in a highly competitive market, exporters must form strategic alliances with producers to obtain the raw material, acquire industrial machinery and distribution logistics equipment, expand storage infrastructure, and manage economies of scale. This includes the procedures to obtain export authorization through the certificates and sanitary documents required by each country [38]. In addition to all of this, the exporters must decide how to process and sell the walnuts. To make a good decision on how to process each AB, exporters have to consider factors such as the size, color, defects, and shape of the walnut, plus the selling price, cost of processing, and capacity constraints of each different product.

MeliFen sells walnut AB in three main categories: (i) hand-cracked shelled walnuts, (ii) machine-cracked shelled walnuts, and (iii) in-shell walnuts (see Figure 1a,b for the products of categories (i), (ii), and (iii), respectively). Each category requires different standards and processing, and there is great operational complexity. Furthermore, an arriving batch may be sold in more than one category. Here, we provide an overview so that the reader can follow the rest of the exposition. For more details, see Liu et al. [39].

### 3.1. Hand-Cracked and Machine-Cracked Shelled Walnuts

As the names indicate, these are walnuts sold without their shells. That condition could make the product more valuable, although prices may vary depending on the exact features of a walnut AB. Generally, hand-cracked shelled walnut is the most premium product because machine cracking tends to scratch the walnut and deteriorates its visual appeal. On the other hand, hand-cracking is a more expensive and time-consuming process as it requires manual labor. Nonetheless, as previously stated, there is a great deal of variety

within each category. The product is generally more valuable if the walnuts are bigger and the color category is lighter.

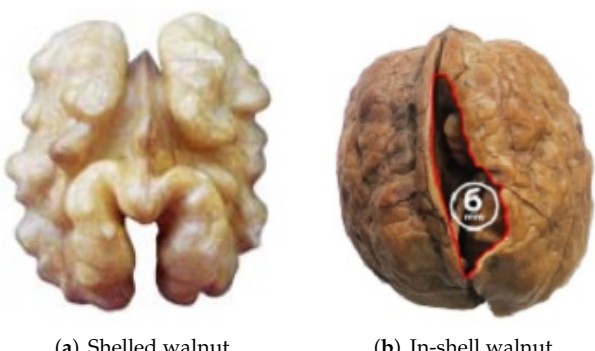

(**a**) Shelled walnut          (**b**) In-shell walnut

**Figure 1.** Main walnut products.

Hand-cracked selling products are: (i) halves, (ii) quarters, and (iii) pieces. The halves product is the highest-value product in this category. It is categorized by size (premium and normal) and color (extra light (EL), light (L), light amber (LA), yellow (Y), and amber (A)). Due to human error, there is a loss in obtaining halves that will be sold as quarters or pieces.

Figure 2 illustrates this behavior, where $W$ represents AB weight (kilogram) (we refer to a specific AB, so the subscript is avoided). Table 1 performs the math for each branch of the flow. The proportion of the halves in AB is $\alpha_{halves}$; $\alpha_{kernel}$ represents the proportion of kernel concerning the total AB weight; $\alpha_{kwd}$ represents the proportion of kernel with no defects. The loss of halves due to manual cracking that can be sold as quarters or pieces is denoted by $\hat{\alpha}_{halves}$. The loss of halves due to manual cracking that cannot be sold as quarters or pieces because they are too damaged is $\hat{\alpha}_{kernel}$. These parameters can be estimated from a AB sample and an exporter's inside knowledge of its operational efficiency.

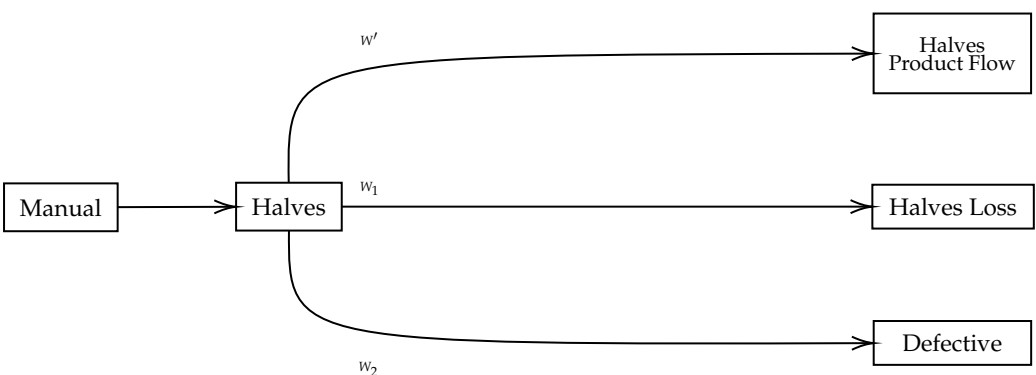

**Figure 2.** Halves by product flow.

**Table 1.** Halves by product flow branch calculations.

| Quantity | Calculation |
|----------|-------------|
| $W'$ | $W \cdot (\alpha_{halves} - \hat{\alpha}_{halves}) \cdot (\alpha_{kernel} - \hat{\alpha}_{kernel}) \cdot \alpha_{kwd}$ |
| $W_1$ | $W \cdot \alpha_{halves} \cdot (\alpha_{kernel} - \hat{\alpha}_{kernel}) \cdot \alpha_{kwd}$ |
| $W_2$ | $W \cdot \alpha_{halves} \cdot (\alpha_{kernel} - \hat{\alpha}_{kernel}) \cdot (1 - \alpha_{kwd})$ |

If we call $W'$ halves that are going to be processed in the *halves product flow*, each type of color's amount is the product between $W'$ and $r_c$, where $r_c$ is the yield for every color $c$. This rule applies except to the EL and L categories, in this case a fraction *tol* of light product that can be sold as an EL.

In Figure 3, there is an additional condition for EL halves. If $r_{EL}$ is less than a business constant (typically 60%, based on an economic balance), all the EL halves will be sold as L halves. Quarters and pieces are only categorized by color. The same colors for halves apply to quarters and pieces, except that EL and L are grouped as one. Sell prices for quarters and pieces are significantly lower than halves. Furthermore, losses due to manipulation in the production line are sold as industrial products.

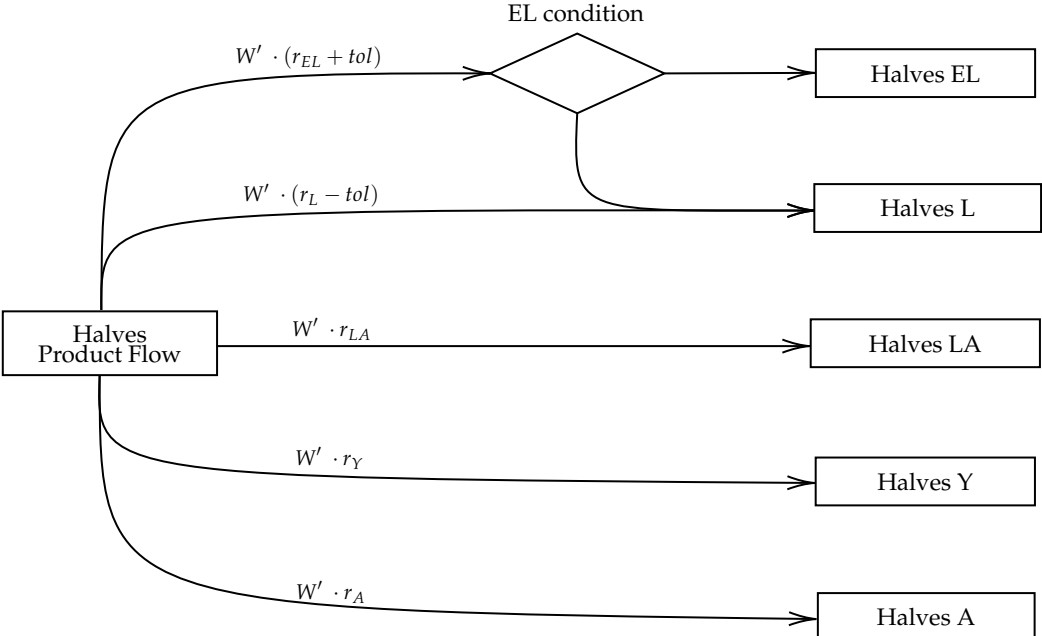

**Figure 3.** Halves products.

Machine-cracked shelled walnut products result in halves, quarters, and pieces. The difference with hand-cracked is the quality since the machine tends to scratch the product. This form of selling may be helpful when the overall quality of the AB is enough for hand-cracked selling.

### 3.2. In-Shell Walnuts

In-shell walnut is walnut sold with its shell. In general, this makes it less valuable on the market than shelled walnut. Nonetheless, it does present some advantages. First, compared to the shelled categories, walnuts do not require as much processing to be sold in-shell. The arriving batch is separated by size because it is the most significant factor that affects the price (while bigger, more expensive; see Table 2). Secondly, almost any walnut can be sold in-shell instead of shelled products, which requires a minimum size to ensure the walnut does not shatter in the process.

**Table 2.** Main size categories for in-shell walnuts. The size refers to the equatorial diameter, as measured with a specialized instrument.

| Trade Name | Diameter $d$ (mm) |
|:---:|:---:|
| 26 | $d < 26$ |
| (26–28) | $26 \leq d < 28$ |
| (28–30) | $28 \leq d < 30$ |
| (30–32) | $30 \leq d < 32$ |
| (32–34) | $32 \leq d < 34$ |
| (34–36) | $34 \leq d < 36$ |
| (+34) | $34 \leq d$ |
| (+36) | $36 \leq d$ |

As can be gathered from our description, it is seldom apparent how an AB should be sold and industrialized. MeliFen has several resources (be it machines or trained manual labor) at its disposal to perform a specific transformation, i.e., size classification devices. The sequence of transformations is called the process, and each step of the process provides semi-finished batches, except the last step, which gives by-product batches.

To industrialize an AB, at least one of the by-product batches should gain value so that the initial value of the AB increases after industrialization. Theoretically, you could apply multiple sequential processes to an AB and gain value from each one. Nevertheless, in practice, MeliFen never performs more than two processes on the same walnut because of operational limitations, so we took the same approach for our formulation.

Figure 4 illustrates the level of complexity that combinations of selling strategies can have. In this case, we considered hand-cracked shelled walnut and in-shell walnut for the same AB. Furthermore, quality checks, business constraints, and tolerances have to be considered in every product branch.

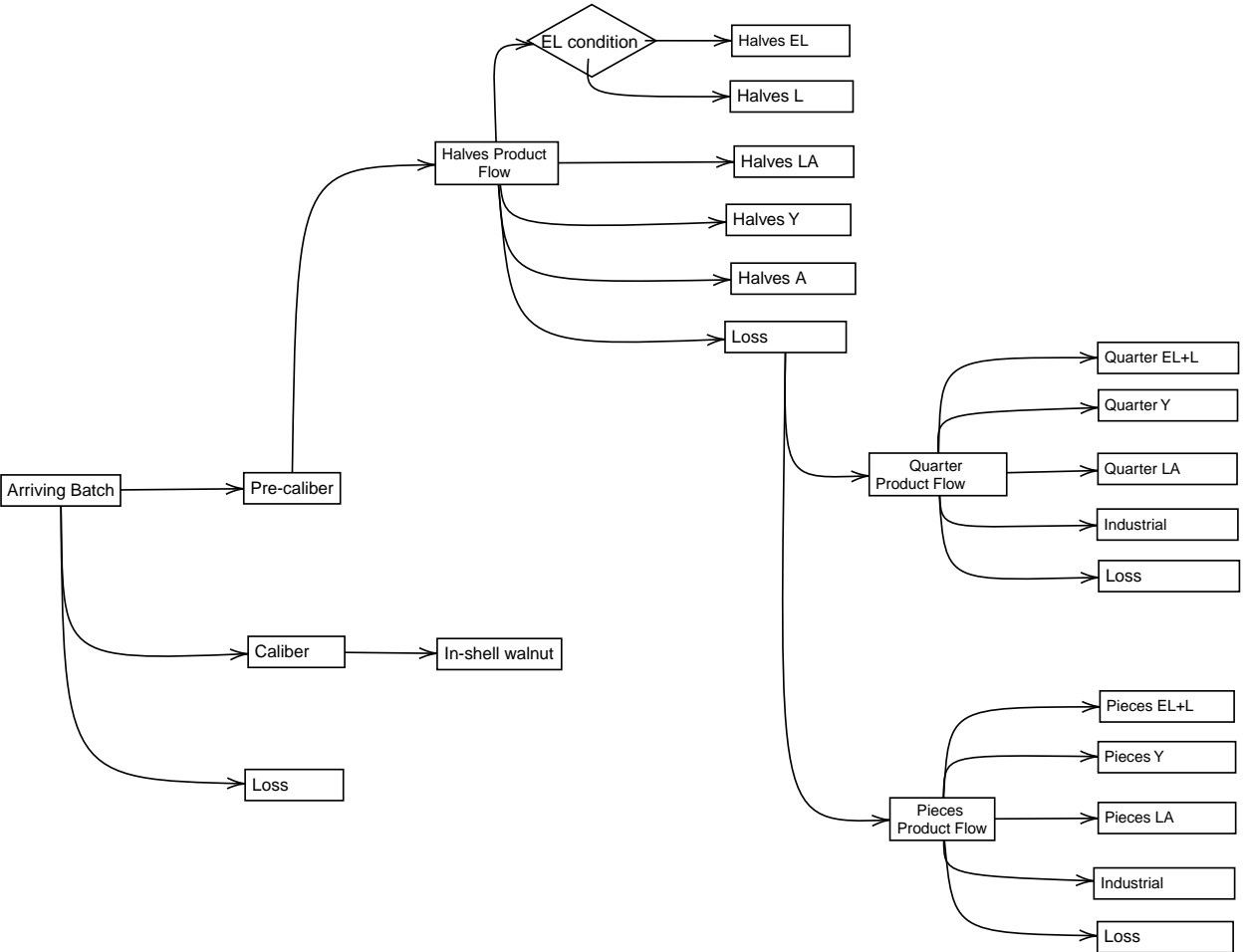

**Figure 4.** Arriving batch selling strategy example.

## 4. Problem Mathematical Formulation

This section introduces the exporter decision model. First, we present the main features of the model. Second, we introduce an alternative solution that we implemented in the DSS. Finally, the mathematical model is shown in detail.

### 4.1. Nature of Exporter Decision-Making

This paper is focused on the exporter decision-making process. In this context, even when decisions should be convenient for him/her, it is necessary to keep the business

sustainable. The exporter's goal is to maximize his/her profit through selling and industrialization services. The producer benefit should also be increased because the exporter's profit depends mainly on this. In this framework, the primary responsibility of the exporter is to negotiate the best possible deal for their producers. However, some exporter decisions could be perverse for the producers, so some constraints were added to the decision exporter scheme to keep the supply chain sustainable.

We treated these topics as follows:

- The exporter offers specific industrialization services to the farmer for an AB $\ell$. The industrialization cost is $c_{I,\ell}$ and the revenue $r_{I,\ell}$. The commission of the commercial operation does not subsidize any industrialization service;
- The goal of the exporter is to maximize his/her own profit in any AB. It is defined as $r_{C,\ell} + r_{I,\ell} - c_{I,\ell}$, where $r_{C,\ell}$ is the commission of the commercial operation, and $r_{I,\ell} - c_{I,\ell} \geq 0$;
- The farmer's final profit of an industrialized AB, $V_{I,\ell}$, is always bigger than the profit of selling the AB without any industrialization, $V_\ell$. Formally, $V_{I,\ell} \geq V_\ell$. The farmer's profit considers all the costs linked to the industrialization and exportation of the walnut, while the income is based on the price of the processed AB;
- As the selling and industrialization decisions may lead to differences between producers with a similar AB, making the sustainability of the supply chain weaker, the exporter's conditions to make the aforementioned decisions are as follows:
  - The planning task is made weekly. A rolling horizon (Sahin et al. [40]) could be more reasonable because it allows frequent information updates. However, part of the decisions made previously can change, meaning that the commercial conditions have been modified. This situation is not desired by the exporter and the producers;
  - The stocks of walnut to be processed in the planning DSS are (1) the available stock of ABs that is not assigned to specific commercial contracts and (2) the ABs that the producers for the planned week commit;
  - If two owners (producers) send ABs with similar features and final profit, the priority will be for the first received in the exporter facility;
  - The DSS updates the final product prices weekly, based on market estimations and historical information;
  - The exporter uses a compensation system to distribute the market-expected price gap among producers. However, this procedure is out of the scope of this work.

The same AB can be processed sequentially, so different quantities could be assigned to different processes. This behavior can easily lead to non-linearity in the optimization model, which can be problematic for a solver. We illustrate this with a simplified scenario, where the goal is to maximize the profit (an oversimplification of the actual exporter goal). On a given $\ell$, we can apply a process $i$ and other $j$. As a result, we end up with two industrial batches:

1. The first industrial batch can be sold as product $i$ (we assumed a one-to-one correspondence between the process applied and the product sold);
2. The second industrial batch can be processed to obtain another product $j$;
3. Following the current industrial policy, the remaining batch that is neither $i$ nor $j$ product is considered a loss.

The respective sizes of the first and second industrial batches depend on the specific characteristics of the AB, the market demands, and the rest of the available raw material. Figure 5 represents this idea. Here, $W_\ell$ denotes the AB weight in kilograms; $k_{\ell,i}$ and $k_{\ell,j}$ indicate the percentage of product types $i$ and $j$ obtained after each process fee has been industrialized, respectively, $i$ and $j$ being two of the process in the set $\mathcal{I}$. Suppose processing costs $C_i$ and $C_j$ (both in USD per kg) and selling prices $p_i$ and $p_j$. Then, this optimization model will look as follows.

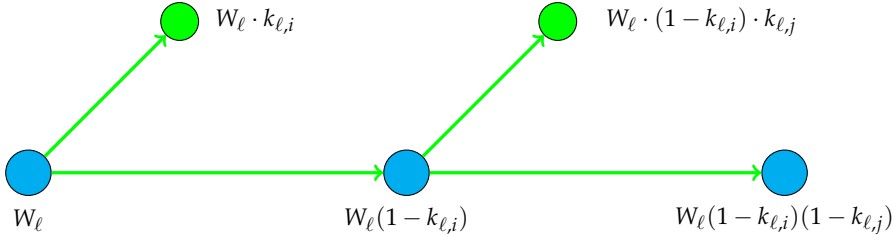

**Figure 5.** Sequential processing of ABs.

The decision variables are $x_{\ell,i}$ and $x_{\ell,j}$, both binary variables where 1 means that the AB was processed by the process $i$ and/or $j$. The set $\mathcal{L}$ belongs to the all arriving batches that are computed in the specific planning instance.

$$\max \sum_{\ell \in \mathcal{L}} \sum_{i \in \mathcal{I}} \sum_{j \in \mathcal{I}} x_{\ell,i} x_{\ell,j} \left( W_\ell k_{\ell,i} p_i + W_\ell (1 - k_{\ell,i}) k_{\ell,j} p_j - W_\ell c_i - W_\ell (1 - k_{\ell,i}) c_j \right) \tag{1}$$

subject to:

$$\sum_{i \in I} x_{\ell,i} = 1 \qquad\qquad \forall \ell \in \mathcal{L}, i \in \mathcal{I} \tag{2}$$

$$\sum_{j \in I} x_{\ell,j} = 1 \qquad\qquad \forall \ell \in \mathcal{L}, j \neq i \in \mathcal{I} \tag{3}$$

$$x_{\ell,i}, x_{\ell,j} \in \{0,1\} \qquad\qquad \forall \ell \in \mathcal{L}, i, j \in \mathcal{I} \tag{4}$$

The objective function (1) sums the sales income and subtracts the cost of processing both arriving batches. As you can see, the objective function includes an evident non-linearity in the form of a product. The presence of a non-linearity makes sense. It stems from the fact that we must combine two decisions: how to process an AB and re-process it to estimate the final income and costs. Constraints (2) and (3) indicate that if you decide to sell an AB as a type-$i$ walnut, you cannot sell the same kilos as another type of walnut $j$. The nature of the decision variables is specified in Constraint (4) (both binary, in this case).

The actual case is even more complex, but this simplified scenario demonstrates that non-linearity occurs naturally in this problem. We introduced a solution strategy that divides the problem into two phases to deal with it. The first phase is a preprocessing that computes the most relevant parameters for the problem, creating a ranking of benefits after an AB is evaluated in an exhaustive list of combinations of processes. The second phase is the proper IP model optimization that determines the best way to sell the walnut lots based on the previous ranking.

*4.2. Solution Strategy: Two-Phase Algorithm*

The sale options for an AB are determined by the processes assigned to it. Even when there are several combinations of processes, Table 3 shows a summary of the possibilities for by-products.

**Table 3.** By-products.

| By-Product | Size | Color | Subproducts |
|---|:---:|:---:|:---:|
| In-shell walnut | ✓ | | ✓ |
| Machine-cracked shelled walnut | | | ✓ |
| Hand-cracked shelled walnut | ✓ | ✓ | ✓ |

The preprocessing phase is dedicated to understanding which conditions are more valuable to an AB. To do that, the exporter should compute the results of different processes and rank the benefits of doing so with the AB.

The first step is to calculate the proportion of the AB that could be used (achieving market quality) to supply the demand. In the case of *hand-cracked shelled walnuts*, halves could be classified by size and color, while quarters and pieces only by color. Considering Section 3, the halves product flow is denoted by $W'$. This fraction of the AB is qualified by the size parameter, so the next step is to classify by color. The mass balance is in Table 4, where each row means the total amount of the specific by-product that the exporter can obtain from an AB denoted by $\beta_i$ when the AB is processed in the $i$ process. In this case, the yields for every color in the specific feed are represented by $r_i$, where $i$ represents one of the colors in which a sized semi-finished product could be classified.

**Table 4.** Types of halves.

| Color | $\beta$ Calculation |
|---|---|
| Extra Light (EL) | $W' \cdot (r_{EL} + tol_\ell)$ |
| Light (L) | $W' \cdot (r_L - tol_\ell)$ |
| Light Amber (LA) | $W' \cdot r_{LA}$ |
| Yellow (Y) | $W' \cdot r_Y$ |
| Amber (A) | $W' \cdot r_A$ |

Following the same steps, fractions were estimated for quarters and pieces in the hand-cracked walnuts shelled, machine-cracked, and in-shell walnuts (more details in Appendix A).

After defining the possible yields in different by-products (for each $\ell$), we defined the revenue for the process pair considering that the exporter only applied two to the same AB. The notation is ($r_{\ell,i,j}$) for tuple (AB, by-product $i$, by-product $j$), and its cost is ($c_{\ell,i,j}$), with $i \neq j$. Both processes are part of a set $\mathcal{I}$, the complete list of available processes in the exporter facility.

The processing tuple replaces the binary variables' products presented before.

$$r_{\ell,i,j} = (\beta_{\ell,i} p_i + \beta_{\ell,j} p_j)C + r_{\ell,i} + r_{\ell,j} \tag{5}$$

$$c_{\ell,i,j} = C_{e,i}\beta_{\ell,i} + C_{e,j}\beta_{\ell,j} + C_i\beta_{\ell,i} + C_j\beta_{\ell,j} \tag{6}$$

Equation (5) sums the commission for selling the AB processed as $i$ and $j$ and the revenue by the industrialization services. The cost Equation (6) takes into account the exportation ($C_{e,i}$ and $C_{e,j}$, both considering USD/kg) and the industrialization ($C_i$ and $C_j$) costs. This balance is written from the exporter's standpoint.

An extra parameter was used to consider the manual capacity. As the labor requires experience, skilled staffing is bound to operational planning. Two binary parameters, $\delta_i$ and $\delta_j$, were used; the value is one if the by-product uses manual capacity and zero otherwise. Hand-cracked shelled walnut processed quantity, in kilograms, is calculated by Equation (7):

$$M_{\ell,i,j} = \beta_{\ell,i} \cdot \delta_i + \beta_{\ell,j} \cdot \delta_j \tag{7}$$

The second phase is the optimization problem. All binary variables represent decisions that indicate how an AB will be sold. It was assumed that preprocessing has been done, so for each AB, the yield, cost, and revenue of each processing option were calculated. Now, we go over the sets, parameters, variables, formulation, and constraint explanations.

The objective function (8) is the sum of benefits obtained if an AB is processed in a specific tuple. Therefore, we created a new set that was composed of the ordered pair $(i, j)$, which represents any feasible combination in set $\mathcal{I}$. The set of ordered pairs is denoted $\mathcal{T}$, and any pair is $t$. There is a special pair that belongs to not processing–not processing, the unique instance when $i$ is similar to $j$.

The following is an explanation of the constraints. Constraint (9) indicates that the revenue for the exporter in the selected tuple for process $\ell$ must be at least equal to the cost

of the industrialization service. Constraint (10) represents the producer's industrialization convenience to process the AB according to the tuple $t$, where the producer's benefit should be better than selling the AB without any processing. The expression $V_{\ell,t}$ is defined as $\beta_{\ell,i}(p_i - C_i - C_{e,i}) + \beta_{\ell,j}(p_j - C_j - C_{e,j})$. The price of the AB without any processing is $p_0$, and the exportation cost, $C_{e,0}$, both in USD/kg. Constraint (11) indicates the fundamental constraint: each AB must be processed according to a tuple. Constraint (12) deals with an operational issue: total manual processing capacity is limited to $D$. Constraint (13) states the nature of the decision variables (binary, in this case).

$$\max \sum_{\ell \in \mathcal{L}} \sum_{t \in \mathcal{T}} x_{\ell,t}\left(r_{\ell,t} - c_{\ell,t}\right) \tag{8}$$

subject to:

$$c_{\ell,t} \leq r_{\ell,t} x_{\ell,t} \qquad \forall \ell \in \mathcal{L} \tag{9}$$

$$W_\ell(p_0 - C_{e,0}) \leq V_{\ell,t} x_{\ell,t} \qquad i,j \in t, \forall \ell \in \mathcal{L} \tag{10}$$

$$\sum_{t \in \mathcal{T}} x_{\ell,t} = 1 \qquad \forall \ell \in \mathcal{L} \tag{11}$$

$$\sum_{\ell \in \mathcal{L}} \sum_{t \in T} M_{\ell,t} x_{\ell,t} \leq D \tag{12}$$

$$x_{\ell,t} \in \{0,1\} \qquad \forall \ell \in \mathcal{L}, t \in \mathcal{T} \tag{13}$$

## 5. Web-Based Integer Programming Decision Support System

Many calculations were required to optimize the processing of walnut ABs due to numerous variables and parameters considered in our problem; therefore, it was necessary to provide a user-friendly practical tool to decide in real-time and, occasionally, with urgency. The web application allows users to enter parameters through a user-friendly interface, facilitating decision-makers to weigh their options before making a final choice.

This section describes some of the implementation details of the optimization solution, as well as the use and operation of the DSS designed to manage the walnut processing.

### 5.1. Implementation of the Optimization Algorithm

The optimization procedure was programmed using Python 3.6. The preprocessing phase primarily employs native Python, as well as Pandas (Version 0.25.3) for straightforward and enhanced data processing capabilities. After preprocessing the AB data, the actual optimization problem is solved using CBC (Version 2.7.5) [41], an open-source mixed-integer programming solver written in C++ and part of the COIN-OR project [42]. We used CBC via PuLP (Version 2.1) [43], a Python package that can serve as an interface for various solvers such as Gurobi and CBC. We chose CBC because it is free and because the optimization problem, thanks to the preprocessing phase, is not so complex that it would require the cutting-edge features that a commercial solver such as Gurobi would provide. The problem is linear (as previously stated) and not too complicated: the number of variables is $|L| \times |P|$, where $|L|$ is the number of ABs and $|P|$ is the number of different ways to sell walnuts. For MeliFen, this corresponds to $224 \times 17 = 3808$, which poses no difficulty to CBC.

The optimization algorithm itself is compatible with both Windows 10 64 bit and UNIX-based operating systems. For a typical use case, it takes around 70 s for the algorithm to perform the preprocessing and then a few extra seconds to find the optimal solution (the exact time can vary depending on the number of ABs being considered). The algorithm is fast enough to try different combinations and conduct sensitivity analysis on the key parameters, perfectly in line with the needs of MeliFen.

The complete DSS encompasses much more than just the algorithm. It consists of a web application, a database, a computing service for the optimization algorithm, and several reporting modules. All these services were deployed on Amazon Web Services (AWS) [44].

Figure 6 shows a diagram of the different services and their relationships. In Section 5.2, we go into more detail about the web application and its usability.

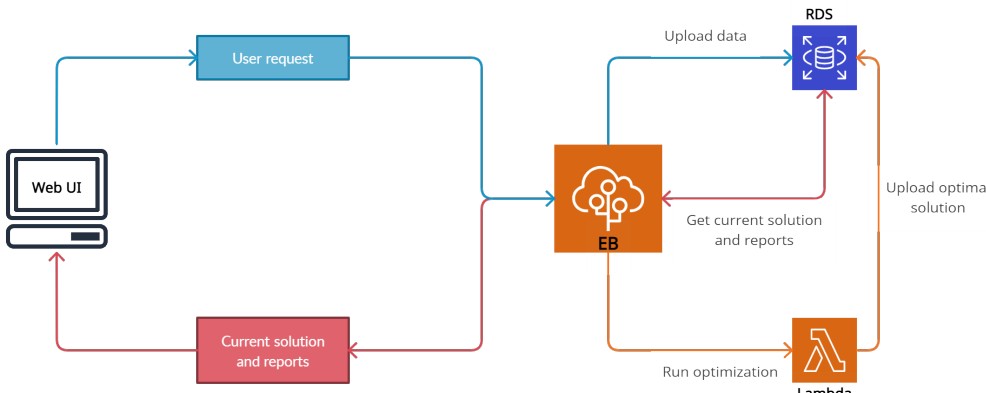

**Figure 6.** AWS architecture for the DSS. It includes the following services: (1) a web application deployed through AWS Elastic Beanstalk (EB), (2) an AWS Lambda function that runs the optimization procedure, and (3) an AWS relational database (RDS) that stores all input and output data. The main functionality for the user consists of (1) uploading data (parameters, values, and AB information) to the database, (2) running the optimization procedure (via a Lambda function that subsequently uploads the optimal solution to the database), and (3) requesting the latest optimal solution and related reporting. All of the requests are made through a user-friendly web interface (Web UI).

## 5.2. Web Application

The web application serves as portal for the decision-maker to interact with the main categories: *general*, *reports*, *spreadsheets*, and *AB optimization*. These categories are composed of several tabs (shown in the first column of Figure 7) through which users can access the following functionalities:

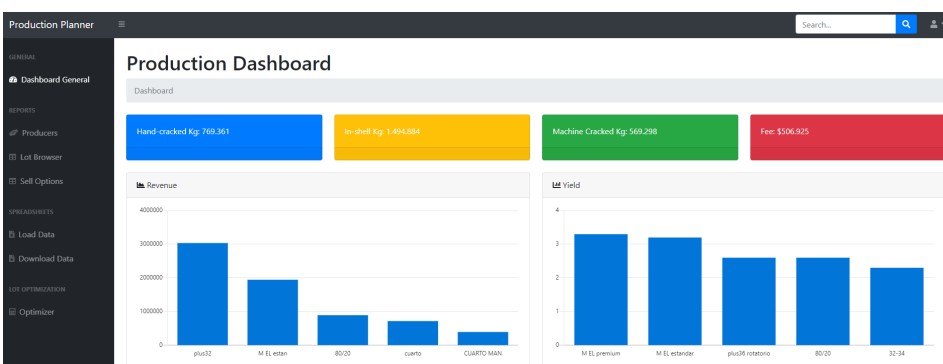

**Figure 7.** General dashboard.

1. Data upload and storage: In the *Load Data* tab of the *spreadsheet* category, users can upload information on the characterization and parameters of the walnut ABs, which are subsequently stored in the database. The data must be edited and uploaded to the web application via two Excel spreadsheets;

    (a) Characterization spreadsheets: The user must register the information of the walnut ABs of each producer, specifying the kilograms of reception, variety, percentages of walnut sizes, percentages without defects, percentages with external defects (imperfect shell, open shell, serious stains, slight stains, split walnut, broken walnut, cracked walnut, adhering hull presence), internal defects (slight shriveling, insect damage, active fungus, inactive fungus, rancidity, severe shriveling, empty walnuts), units per kilo, and percentages of the AB

composition (shape, color, shell, kernel), according to the Chilean Walnut Commission (ChWC) [45]. It should be noted that part of the information is obtained from a representative sample of each AB;

(b) General parameters spreadsheets: The user must register a series of parameters standardized by the ChWC [45] to comply with the quality requirements for exporting walnuts. These parameters are related to the percentage of walnut defect tolerances and yields of walnut ABs according to variety and shape. In addition, they must complete other company-specific parameters focused on the price of the different walnut varieties (category, type, color, size), losses and processing costs of the walnuts, export costs per kilo of walnut (in-shell walnuts and shelled walnuts), process capacities (machine cracked, hand-cracked, separator machine), and the exporter fee. Details can be found in Appendix A;

2. Optimization: Once the data have been loaded, the customer can run the optimization procedure in the *Optimizer* tab of the *AB optimization* category. By clicking on the *Execute* button, the optimization algorithm is displayed as a Lambda function running on the Amazon Linux distribution. Subsequently, it loads the optimal solution into the database. The database stores parameters, AB information, and optimal solutions and includes versioning;

3. Reports: The reports are mainly calculated in the back-end of the web application and displayed in the front-end as different dashboards with the KPIs detailed below;

(a) General Dashboard tab: The user is provided with production information on the number of kilograms of in-shell walnuts and shelled walnuts (hand-cracked and machine-cracked) that would be produced if the walnut ABs were processed as indicated in the optimal solution. Furthermore, it provides the total fee that the exporter would obtain, as well as a graph of revenues and yields by walnut classification and color, respectively (see Figure 7). In addition, it presents a summary table by product (see Figure 8) that is divided into six columns: macro category (shape of walnut ABs processing), category (size, color, shape), revenue, production costs, export costs, and kilograms sold;

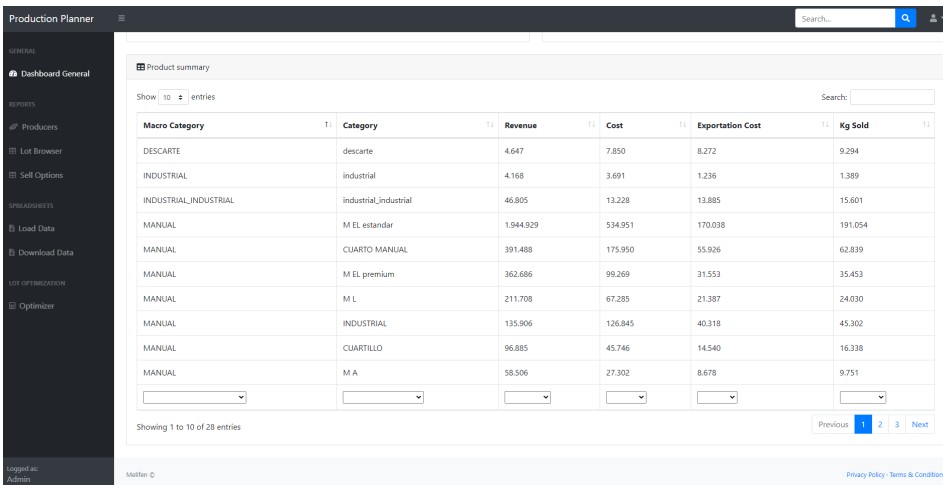

**Figure 8.** Dashboard summary by product.

(b) Producers tab: This presents a graph of each walnut producer as a function of the profitability of kilograms optimally processed (see Figure 9). This tab also includes a summary table for each producer, subdivided into 12 columns: producer, revenue, production costs, export costs, exporter fee, total kilograms, total yield, and yield per kilogram of each variety and kernel of the walnut;

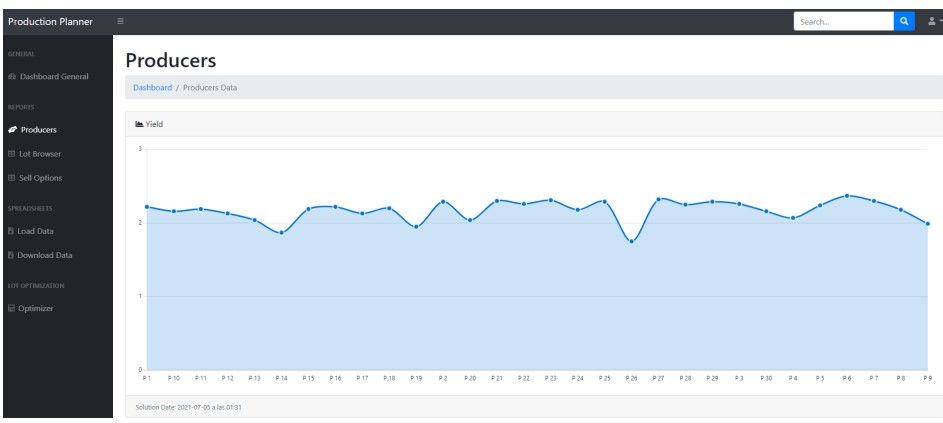

**Figure 9.** Profitability per kilogram of each producer.

(c)  AB Browser tab: This presents a table that summarizes the optimal processes performed on each AB by walnut producer, which is subdivided into the same columns as the *General Dashboard*, but includes the following additional columns: AB number, walnut variety, producer, and kilograms processed;

(d)  Sell Options tab: This presents a table that summarizes the sales options for each AB, subdivided into nine columns viz. AB number, processing order, sales detail (forms of ABs processing), revenue, production costs, export costs, margin, exporter fee, and return to the walnut producer.

In addition, the user can request the most recent optimal solution and related reports in the *Download Data* tab. The results can be exported as an Excel spreadsheet, allowing decision-makers to sensitize the model parameters in a reasonable computational time.

## 6. Computational Results

This section estimates the potential benefits of incorporating a DSS into the walnut AB processing optimization. We present a comparison of the results obtained from the DSS applications, which take into account all operational constraints, versus the company's manual AB processing. To do so, we used the company's 2019 demand data.

### 6.1. AB Demand Characterization

We tested the DSS using the actual demand MeliFen faced in the year 2019. The demand was 224 ABs, totaling almost 2500 tons of product to be processed by 30 nut producers. MeliFen estimates AB characterization such as size, internal and external defects, and product yields and provides them as input to the DSS.

### 6.2. Performance Metrics

To estimate the benefits of using the DSS in AB processing, we compared MeliFen's manual processing decisions to the optimized solution of our model. Each solution is defined as an AB-specific tag that indicates the processing decision. We computed the revenue, costs, and amount of product generated for each AB decision, which were used to measure the following performance metrics:

- Total revenue in USD;
- Total cost in USD;
- Producers profit per kilogram (USD/kg);
- Amount of product (kg).

### 6.3. Experiment Results

Our baseline experiment considered 224 ABs that summed up to almost 2500 t of fruit. Each AB has 17 possible forms of processing options, and the objective was to choose the one that maximizes the producers' profit. Table 5 shows the tonnage processed, revenue,

costs, and fee for the category of by-products. The fixed solution processed 51% more hand-cracked product than the optimized planning, to the point where a fixed solution was theoretically infeasible due to the hand-cracked capacity of 1,000,000 kg.

Moreover, the optimized planning processed more product in all categories except hand-cracked. The total economic impact is presented in Table 6. Overall, the results indicated that the optimized planning generated almost the same revenue as the fixed plan, with a 0.1% difference. In terms of costs, the optimized planning reduced the AB processing costs by 16.6%. Overall, the DSS generated 9.8% more profit than the fixed plan, while the company reduced its commission by only 0.1%.

**Table 5.** DSS versus fixed solution metrics benchmark.

| Product | Metrics | | | |
|---|---|---|---|---|
| | Ton Proc. | Revenue (*k*USD) | Cost (*k*USD) | Fee (*k*USD) |
| Hand-cracked diff. (%) | 51 | 53 | 56 | 53 |
| Machine-cracked diff. (%) | −40 | −40 | −33 | −40 |
| In-shell diff. (%) | −18 | −28 | −26 | −28 |
| Waste diff. (%) | −44 | −46 | −36 | −46 |
| Fixed hand-cracked | 1164 | 4950 | 2251 | 297 |
| Fixed machine-cracked | 343 | 1103 | 330 | 66 |
| Fixed in-shell | 1225 | 2378 | 397 | 143 |
| Fixed waste | 29 | 30 | 16 | 2 |
| Optimized hand-cracked | 769 | 3240 | 1443 | 194 |
| Optimized machine-cracked | 569 | 1836 | 494 | 110 |
| Optimized in-shell | 1495 | 3318 | 536 | 199 |
| Optimized waste | 53 | 56 | 25 | 3 |

**Table 6.** DSS versus fixed plan general results.

| Model | Metrics | | | |
|---|---|---|---|---|
| | Revenue (*k*USD) | Cost (*k*USD) | Fee (*k*USD) | Profit (*k*USD) |
| Benchmark diff. (%) | −0.1 | −16.6 | −0.1 | 9.8 |
| Fixed planning | 8461 | 2994 | 508 | 4959 |
| Optimized planning | 8449 | 2498 | 507 | 5444 |

We compared the yields of the producers in both solutions to analyze the benefits of optimized planning in greater depth. As shown in Figure 10, the optimized planning producers yields were greater than or equal to the fixed planning yield. Table 7 shows the maximum and average difference in yield by the producer. For some producers, the benefits from optimized planning increased their yield by 34, while the average increase in yield reached 12%.

Furthermore, the presented fixed plan is impractical because it surpassed hand-cracked capacity and some quality tolerances in ABs. For this reason, the theoretical gap between the optimized and fixed plans was even wider.

**Table 7.** Producers yield benefits from optimized planning.

| | Value (%) |
|---|---|
| Max | 34 |
| Average | 12 |
| Standard deviation | 9 |

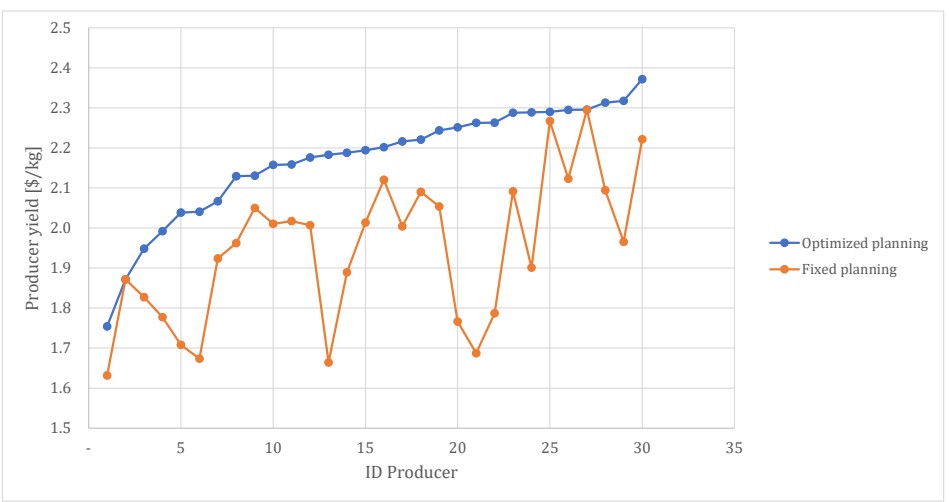

**Figure 10.** Producers' yield benchmark.

### 6.4. Sensitivity Analysis over Hand-Cracked Capacity

Now, we studied the profit sensitivity over different hand-cracked capacity settings. We considered three scenarios in which hand-cracked capacity was reduced by 50% (A), 75% (B), and 100% (C). Table 8 presents the tonnage of processed by-product for each case. The base case with no capacity reduction is represented by (O), and its value is 1000 t.

Since the hand-cracked capacity constraint was active in all cases, ABs were processed in alternative ways. Since in-shell sales are limited by the physical quality of the product (with 20% growth in Case (C)), machine-cracked was the product with the highest growth (86% in Case (C)). Wastes also increased as the machine-cracked category generates the most waste. Detailed profit results are presented in Table 9.

When the hand-cracked capacity constraint was set to zero, general profit dropped by only 2.3%. Nonetheless, the company fee was penalized at 11.6% because it is based on revenue. Even though revenue diminished with a lower hand-cracked capacity, the DSS managed to reduce the costs, making the general profit as stable as possible.

**Table 8.** Production levels for each hand-cracked capacity case.

| Product | Ton Processed | | | |
|---|---|---|---|---|
| | **O** | **A** | **B** | **C** |
| Hand-cracked diff. (%) | - | −35 | −67 | −100 |
| Machine-cracked diff. (%) | - | 21 | 49 | 86 |
| In-shell diff. (%) | - | 10 | 18 | 20 |
| Waste diff. (%) | - | 17 | 36 | 81 |
| Hand-cracked | 769 | 500 | 250 | - |
| Machine-cracked | 569 | 689 | 849 | 1057 |
| In-shell | 1495 | 1648 | 1759 | 1794 |
| Waste | 53 | 62 | 72 | 96 |

**Table 9.** General results for each hand-cracked capacity case.

| Case | Metrics | | | |
|---|---|---|---|---|
| | **Revenue (kUSD)** | **Cost (kUSD)** | **Fee (kUSD)** | **Profit (kUSD)** |
| O: Base case | 8449 | 2498 | 507 | 5444 |
| A: 50% reduction | 8123 | 2225 | 487 | 5411 |
| B: 75% reduction | 7798 | 1945 | 468 | 5385 |
| C: 100% reduction | 7473 | 1705 | 448 | 5320 |

Figure 11 depicts the yield reduction faced by producers as a result of lower hand-cracked capacity compared to the base case. We observed that half of the producers faced a decrease in yield, with a maximum reduction of 17% in the case of zero hand-cracked capacity. This experiment demonstrated that, even if the total profit is similar, some producers can be seriously affected when hand-cracked capacity is reduced.

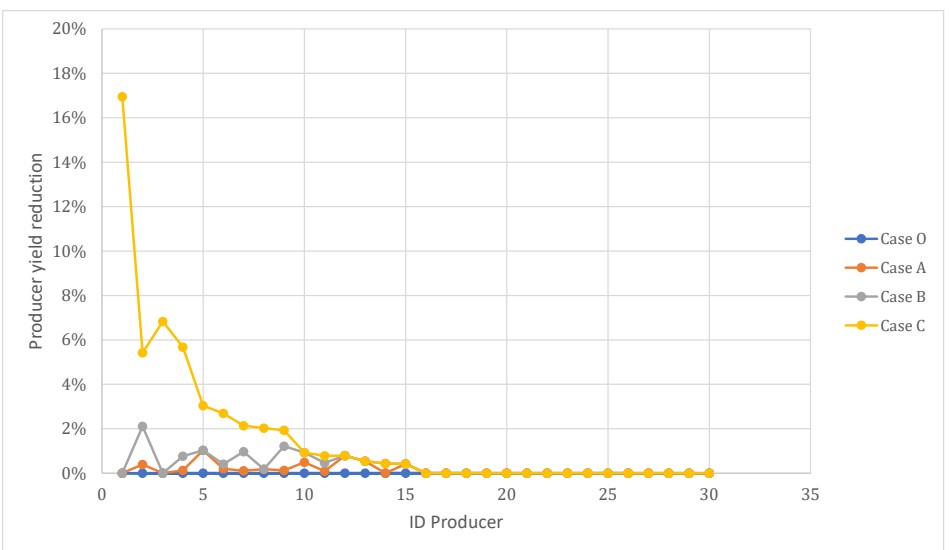

**Figure 11.** Producers' yield reduction sensitivity.

## 7. Conclusions

This study presented an optimization model for the AB processing planning that a walnut exporter faces. Our model considered detailed information from the fruit, such as size, color, and defects, among others. Our model was embedded in a web application, resulting in a decision support system.

In our computational experiments, we obtained profit improvements of 9.8%. In some cases, producers improved their profit by 34%. From the exporter's perspective, our solution sacrificed 0.1% of their fees. This is a small quantity for the benefits obtained. Furthermore, the exporter may charge a greater price to the producers due to better planning.

This study showed important benefits for producers and exporters by using a DSS for production planning. A fixed plan may be biased due to the complexity of multiple forms of processing. In some cases, delivering an infeasible plan may incur higher costs for exporters and producers. Visibility is another benefit since producers can check in real-time how their product is being handled and the expected profit they will receive.

Our model may also be used for strategic decisions, such as the economic evaluation of new machines that may be more cost efficient or machines that could improve the quality of the product. New AB processing options could also be economically evaluated, including a different hand labor type.

Strengthening the link between business and the local communities allows MeliFen to ensure continuity and expansion outward, qualifying it as a global company with the option of notable credits. In this way, the proposed decision support system favors a better diversification of the work, improving the balance and sustainability of the business, which positively impacts the producers' economy and the fairness perception by producers.

Valuable extensions for future research are opened from the current one. For instance, a robust optimization approach could include stochastic sell prices in the model. Other studies may focus on sampling the quality of the AB, including, for example, the detection of foreign objects by computer vision and the segmentation of walnut images through deep learning [46] or machine learning [47] techniques. This could be a major improvement of our model since the quality of the fruit is a crucial input.

**Author Contributions:** Conceptualization, C.F.B.-P., L.A.C.-R. and E.L.A.; methodology, C.A.M.-C.; software, C.F.B.-P. and L.A.C.-R.; validation, T.C.-V.; formal analysis, C.F.B.-P.; investigation, B.A.U.-C. and T.C.-V.; resources, T.C.-V.; data curation, B.A.U.-C.; writing—original draft preparation, C.F.B.-P., L.A.C.-R. and C.A.M.-C.; writing—review and editing, E.L.A. and B.A.U.-C.; visualization, B.A.U.-C.; supervision, C.A.M.-C.; project administration, C.A.M.-C. All authors have read and agreed to the published version of the manuscript.

**Funding:** This research received no external funding.

**Institutional Review Board Statement:** Not applicable.

**Informed Consent Statement:** Not applicable.

**Data Availability Statement:** The data are confidential (protected) due to contracts between MeliFen and the walnut producers.

**Acknowledgments:** The authors would like to thank MeliFen for providing their collected data and for their participation in the case study. C. Monardes-Concha gratefully acknowledges the support from CONICYT-Chile, through CONICYT-PFCHA/Doctorado Nacional/2018-21181771. A. Croquevielle-Rendic acknowledges the support from CONICYT-Chile, through ANID BECAS/MAGISTER NACIONAL 2017-22171763.

**Conflicts of Interest:** The authors declare no conflict of interest.

## Abbreviations

The following abbreviations are used in this manuscript:

| | |
|---|---|
| DSS | Decision support system |
| AB | Arriving batch |
| IP | Integer programming |
| KPIs | Key performance indicators |
| aDSS | Agricultural decision support system |
| MILP | Mixed-integer linear programming |
| EL | Extra Light |
| L | Light |
| LA | Light amber |
| Y | Yellow |
| A | Amber |
| AWS | Amazon Web Services |
| ChWC | Chilean Walnut Commission |

## Appendix A. Calculating $\beta$

*Appendix A.1. Quarter*

The quarters are the second-highest value product from the hand-cracked shelled walnuts. They are priced according to color, with no size category. The colors are extra light (EL) and light (L); yellow (Y); and light amber (LA). Other colors are sold as an industrial product. Similar to the halves, the most expensive colors are the first two, EL and L. In contrast, LA is the least-expensive color.

As the halves case, when the walnut is opened, there is a loss due to the process that cannot be sold. The flawed fruit is sold as an industrial product. Figure A1 illustrates the product flow.

Figure A1 considers an arriving batch (AB) of size $W'$ with processing loss taken into account. The amount of product by color is given by the yield $r_i$, where $i$ identifies one of the processes. Table A1 shows the calculation for each by-product and the average price per kilogram. The industrial by-product is given by all products not defined in Table A1.

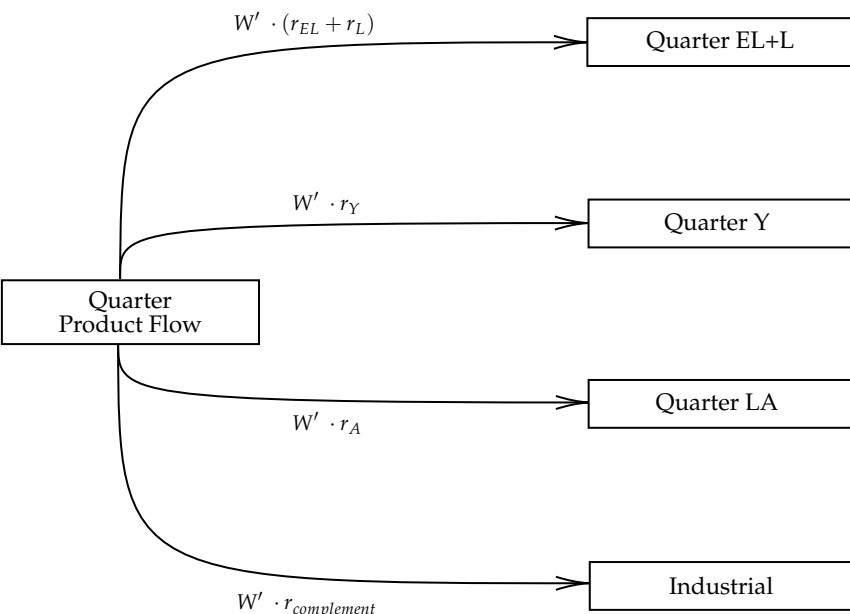

**Figure A1.** Quarter products.

**Table A1.** Types of quarters.

| Color | $\beta$ Calculation |
|---|---|
| Extra Light and Light | $W' \cdot (r_{EL} + r_L)$ |
| Light Amber | $W' \cdot r_{LA}$ |
| Yellow | $W' \cdot r_Y$ |

*Appendix A.2. Pieces*

The final by-product that can be sold as hand-cracked is pieces. It is also priced according to the same color categories as the quarters. Table A2 contains the price by color for each piece by-product.

**Table A2.** Types of pieces.

| Color | $\beta$ Calculation |
|---|---|
| Extra Light and Light | $W'_\ell \cdot (r_{EL} + r_L)$ |
| Light Amber | $W'_\ell \cdot r_{LA}$ |
| Yellow | $W'_\ell \cdot r_Y$ |

*Appendix A.3. Machine-Cracked Shelled Walnut*

This product is machine-made, resulting in a lower-quality product than hand-cracked. This process generates four by-products: 80/20 (a mixture between halves and quarters), quarters, pieces, and industrial.

This process is used when an AB cannot be sold reasonably as hand-cracked or in-shell. Generally, this process is applied for the worst-quality part of an AB or the entire AB that cannot be sold in other formats.

The by-product yield depends on whether the whole AB has been processed in the machine or just a portion of it. This yield is an empirical value that is applied to each AB. Table A3 presents the yield by-product.

**Table A3.** By-product machine-cracked.

| By-Product | $\beta$ (Whole Arriving Batch) | $\beta$ (Portion) |
|---|---|---|
| 80/20 | 60% | 35% |
| Quarters | 25% | 50% |
| Pieces | 15% | 10% |
| Industrial | 0% | 5% |

*Appendix A.4. In-Shell Walnuts*

Walnuts can also be sold in-shell. Sales prices for in-shell walnuts are usually much lower than shelled walnuts (especially hand-cracked), but their processing is straightforward and less time-consuming. Furthermore, not every AB can be sold shelled (at least not for a reasonable price) because strict standards regarding the quality and number of defects in the AB must be met. There are strict standards for in-shell walnuts and different price categories based on walnut size, quality, and defects, and thus, the best sales option will ultimately depend on the quality of each AB.

For in-shell walnuts, the most significant factor affecting the price is the size of the walnuts. Table 2 lists the main categories of in-shell walnuts according to size. The bigger the walnut, the higher the selling price. The standards allow for some flexibility: many walnuts can be sold in any given size category, provided that at most, 10% of the AB is smaller than that category.

According to Table 2 , the product is sorted by size. Between the diameters of 26 mm and 36 mm, there are five ranges, for example 26–28 and 28–30. The product with a diameter greater than 36 mm is an additional category, as is a product with a diameter less than 26 mm.

The fact that an AB can have at most 10% of the walnuts smaller than the category's AB opens up the possibility of interesting optimization questions. Suppose an AB contains 10% walnuts in the category (32–34), 80% in the category (34–36), and 10% in the category +34. You could sell the entire AB as +34, which is simple from an operational perspective since you do not have to separate the AB. On the other hand, you could obtain a better value by selling the +36 part as +36 and the rest as +34. Is the additional profit greater than the cost of separating the AB by size? That is where the optimization potential lies.

Hence, obtaining the maximum value out of selling an AB of in-shell walnuts requires separating the walnuts by size in an optimal manner. In the case of MeliFen, the separation is performed by the machine, in a process detailed in Figure A2. Basically, the machine separates the walnuts into two groups called *caliber* and *pre-caliber* at a specific cut-off diameter ($d$). For $d = 34$, the caliber would be the group corresponding to +34 ($d \geq 34$), while the pre-caliber would be the group of smaller walnuts. Furthermore, separation results in a *loss* percentage: walnuts that were not sorted into any of the groups. Both the pre-caliber and the loss are typically machine-shelled and sold in this manner.

The *loss* group does not correspond to the actual loss, but rather to walnuts that were not classified yet, but can be used (usually sold as machine-shelled). On the other hand, the caliber group needs to go through one more step. Only a certain proportion *tol* of the AB meets the required standards for in-shell sale. Hence, only a *tol* fraction of the caliber group is sold in-shell, with the remainder typically sold as machine-shelled. This last process is represented in Figure A2.

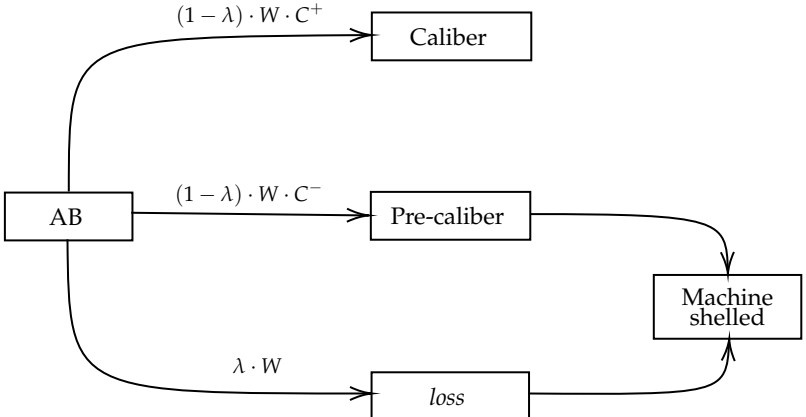

**Figure A2.** Separation of an AB by walnut size, showing how many kilograms of walnut go into each group. In the diagram, $\lambda$ represents the *loss* percentage of the process (a parameter of the separator machine), $W$ is the total kilograms in the AB, $C^-$ the proportion of pre-caliber walnuts, and $C^+$ the proportion of caliber walnuts.

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
