# Peer review of "Web-Based Integer Programming Decision Support System for Walnut Processing Planning: The MeliFen Case"

_agriculture, doi:10.3390/agriculture12030430_

Round 1

Reviewer 1 Report

The article is interesting and relevant. Its layout and evaluation of the methodological part show a rather high level. The literature review is comprehensive and the findings are presented in an orderly manner. 

Author Response

Dear reviewer 1:
Thanks for your comments. We have carefully revised the manuscript, fixing the mistakes, typos, and minor spell check. The changes are highlighted with red color, and they are the following ones:

  • Line 157 Change "Zhai et al. (2020)" for "Zhai et al. [15]".
  • Lines 547 to 551, we add a paragraph with policy implications related to the research findings as reviewer 3 required.
  • Line 620 Change "Table ??" for "Table 2".
  • Line 624 Change "According to the table 2" for "According to Table 2".

We hope that changes fulfill your requirements.

The authors.

Reviewer 2 Report

The submitted paper can, in my opinion, be published without any changes.

Author Response

Dear reviewer #1:
Thanks for your comments. We have carefully revised the manuscript, fixing the mistakes, typos, and minor spell check. The changes are highlighted with red color, and they are the following ones:

  • Line 157 Change "Zhai et al. (2020)" for "Zhai et al. [15]".
  • Lines 547 to 551, we add a paragraph with policy implications related to the research findings as reviewer 3 required.
  • Line 620 Change "Table ??" for "Table 2".
  • Line 624 Change "According to the table 2" for "According to Table 2".

We hope that changes fulfill your requirements.

The authors.

Reviewer 3 Report

The paper provides results of an integer programming model on walnut processing planning. Specifically, the authors estimate production and economc effects derived from introduction of decision suppost system in Chile exporters of walnut. The paper is much original and the methodology used is well suitable and applied to the data.
I suggest only few minor revisions in order to improve the quality of the manuscript:
- Table 6. Please, show also the standard deviations
- Conclusions. I suggest to improve discussion about policy implications related to the research findings. This is an important point that is scarcely discussed in the paper

Author Response

Dear reviewer #3:
Thank you very much for your valuable comments on improving the article. Table 6 does not have a standard deviation because we compare only two values. This comparison is between the actual planning of the company versus the optimized plan for the same year. For this reason, the use of standard deviation is not necessary. For Table 7, the analysis is performed for every producer, and the standard deviation is included. We have carefully revised the manuscript, fixing the mistakes, typos, and minor spell check. The changes are highlighted with red color, and they are the following ones:

  • Line 157 Change "Zhai et al. (2020)" for "Zhai et al. [15]".
  • Lines 547 to 551 add a paragraph with policy implications related to the research findings as you requested.
  • Line 620 Change "Table ??" for "Table 2".
  • Line 624 Change "According to the table 2" for "According to Table 2".

We hope that changes fulfill your requirements.

The authors.